# Influence of Selective Laser Melting Technology Process Parameters on Porosity and Hardness of AISI H13 Tool Steel: Statistical Approach

**DOI:** 10.3390/ma14206052

**Published:** 2021-10-13

**Authors:** Filip Véle, Michal Ackermann, Václav Bittner, Jiří Šafka

**Affiliations:** 1Faculty of Mechanical Engineering, Technical University of Liberec, Studentská 1402/2, 461 17 Liberec, Czech Republic; 2The Institute for Nanomaterials, Advanced Technologies and Innovation, Technical University of Liberec, Studentská 1402/2, 461 17 Liberec, Czech Republic; michal.ackermann@tul.cz (M.A.); jiri.safka@tul.cz (J.Š.); 3Faculty of Science, Humanities and Education, Technical University of Liberec, Studentská 1402/2, 461 17 Liberec, Czech Republic; vaclav.bittner@tul.cz

**Keywords:** SLM, Selective Laser Melting, H13, tool steel, additive manufacturing, DOE

## Abstract

The correct setting of laser beam parameters and scanning strategy for Selective Laser Melting (SLM) technology is a demanding process. Usually, numerous experimental procedures must be taken before the final strategy can be applied. The presented work deals with SLM technology and the impact of its technological parameters on the porosity and hardness of AISI H13 tool steel. In this study, we attempted to map the dependency of porosity and hardness of the tested tool steel on a broad spectrum of scanning speed—laser power combinations. Cubic samples were fabricated under parameters defined by full factorial DOE, and metallurgic specimens were prepared for measurement of the two studied quantities. The gathered data were finally analyzed, and phenomenological models were proposed. Analysis of the data revealed a minimal energy density of 100.3 J/mm${}^{3}$ was needed to obtain a dense structure with a satisfactory hardness level. Apart from this, the model may be used for approximation of non-tested combinations of input parameters.

## 1. Introduction

Selective Laser Melting (SLM) is an emerging Laser Powder Bed Fusion (L-PBF) additive manufacturing (AM) process for producing prototypes and fully functional metallic products in a short period of time [1]. Similar to other AM techniques, the SLM technology enables the production of parts that cannot be manufactured by conventional technologies such as machining and pressure casting. Specifically, it is possible to manufacture parts with curved internal cooling channels, lattice structures, and complex geometry shapes [2,3]. The SLM technology also reduces the number of parts in assemblies to one complex design with all the necessary features and proper functions [4]. Another benefit of this technology is that many metallic powders are available on today’s market.

As soon as a new type of metallic powder is developed for the SLM technology in terms of a suitable particle size distribution and other qualitative measures, the technological parameters for reliable melting need to be evaluated [5]. For most applications, the aim is to produce a material with the lowest porosity. Only then can the best mechanical properties be expected for both static and dynamic loading [6,7]. Despite the fact that many numerical tools for this stage of the SLM process were introduced in recent years, the main part still remains purely experimental [8]. The process of finding process parameters usually starts with single line welds under different combinations of laser power and scanning speed with a selected layer thickness. Subsequently, the weld thickness, surface quality, and shape of cross-section cuts are analysed, and the most suitable combinations of parameters are chosen. The second stage of finding the process parameters is focused on printing specimens with a defined volume. During this stage, the main process parameters (e.g., hatch distance and scanning strategy) are evaluated in order to obtain a product with required output parameters [9]. Due to the fact that the porosity of SLM–printed specimens must be verified by either metallographic analysis or CT scanning, the given procedure is highly time- and cost-consuming. With this in mind, any tool that may facilitate the evaluation of the technological process is beneficial.

This article describes a detailed study of SLM process parameters and their influence on the porosity and hardness of the final product. According to the available literature, more than 50 parameters influencing the SLM process may be identified [10]. Using statistical tools, i.e., Full Factorial Design of Experiment (DOE) followed by an evaluation of the gathered data, we designed a mathematical model to illustrate the relation between the selected input and output parameters. The authors believe that such a model may be very useful in finding the technological parameters of SLM process.

The use of DOE for finding SLM technology process parameters is described in several publications. Laakso et al. [11] used a D-optimal DOE approach with H13 tool steel as a reference material. The final porosity of cubic specimens printed with various combinations of scanning speed-hatch distance and laser power-scanning speed was displayed with the use of contour plots. Given study reported that a volume energy density of 100 J/mm${}^{3}$ is needed to obtain a dense structure. Liao et al. [12] used DOE and subsequent Analysis of variance (ANOVA) to quantify of process parameter settings for a binary Ni-Fe alloy. Once again, the study showed significance levels of individual parameters in terms of specimen porosity. Moreover, the combination with the lowest porosity was proposed.

AISI H13 (DIN 1.2344) tool steel was selected as a reference material for this work. This type of steel is frequently used for the fabrication of hot working tools and dies, and in the injection moulding industry. Today, this type of powder is offered as standard by many manufacturers such as SLM Solutions Group AG (Lübeck, Germany), Carpenter Additive (Carpenter Technology Corporation, USA) and Oerlikon Metco (Pfäffikon, Switzerland). Recent studies provide information about the mechanical properties of printed parts and applied process parameters with a wide range of final porosities of parts. The influence of heat treatment on microstructure and mechanical properties has also been investigated for H13 tool steel. B. Vrancken [13] dealt with the influence of preheating on residual stresses. R. Mertens [14] amplified Vrancken’s findings, and mentions that residual stresses are weaker with increasing preheating temperature. J. Krell [15] also dealt with the influence of surface preheating to reducing the possibility to initiate cracks. J. Yan [16] focused on the cooling speed critical for creating a martensitic structure.

At this point it must be stated that our work does not deal with analysis and explanation of defect formation in the volume of specimens built under selected SLM process parameters. The SLM process is observed here as a closed system with its input and output parameters for which we want to find relationships supported by statistics. Despite this fact, we believe it is necessary to introduce the effects which lead to increased porosity. The current literature usually mention three mechanisms which form voids inside the part. Most recognised are the balling effect [17], keyhole effect [18] and a lack of fusion [19]. Each of these mechanisms are caused by different combination of process parameters and they form a defect with distinct shapes and other defining measures.

## 2. Materials and Methods

This article focuses on finding the parameters of the SLM process for H13 tool steel procured from the powder producer SLM Solutions Group AG (Lübeck, Germany). The internal porosity and hardness were chosen as the monitored outputs of the set process parameters. The Full Factorial method of DOE was used for preparing experiment specimens. Experiments were performed on testing cubes with a 10 mm edge length. The initial process parameters were chosen based on the published articles. All of the combinations of the selected parameters produced solid cubes with different porosity values.

### 2.1. Selective Laser Melting

The principle of the SLM technology is based on the use of a laser beam to transfer energy to the preheated metal powder. The manufacturing process takes place inside a closed chamber under a nitrogen (N${}_{2}$) or argon (Ar) shielding atmosphere. The principle of the SLM technology is shown in Figure 1. The process starts with the distribution of powder in a defined layer thickness on the build platform. Then, the laser scans a cross-section of the parts, and the build platform is lowered by one layer thickness. The powder is repeatedly spread and scanned until the part is fully manufactured. After finishing the last layer, the part and the surrounding powder are cooled to stabilise the structure and prevent the initiation of cracks in the parts. Some materials used in the SLM process require heat treatment after manufacturing to stabilise the internal structure and change the mechanical properties [13]. In our case, the H13 was tested in an as-built condition, i.e., with no subsequent heat treatment.

### 2.2. Process Parameters

In the SLM process, it is possible to influence more than fifty process parameters [10]. However, only a fraction of these parameters is usually changed in order to find the process window of the parameters that may create an object with a defined porosity and structure. This process window has to be found for each material [20]. The most influenced parameters are:Laser power *P* [W]Hatch distance *h* [mm]Scanning speed *v* [mm/s]Layer thickness *t* [mm]

These parameters are included in the Volumetric Energy Density VED [J/mm${}^{3}$] equation. This equation is used for a fast comparison of process parameters and is widely used in studies focusing on metal powder-based technologies [21]. VED is based on the fact that each element dV [mm${}^{3}$] absorbs energy dW [J] (Equation (Equation 1)). Adapting this with the use of laser power *P* [W], the element of time dτ [s] and infinitesimal laser beam displacement dl [mm], we obtain the final form of the VED [22] which is frequently used in the field of L-PBF technologies.
(1)$$VED=\frac{dW}{dV}=\frac{P\;\cdot \;d\tau }{h\;\cdot \;t\;\cdot \;dl}=\frac{P}{h\;\cdot \;t\;\cdot \;v}$$

For each type of powder, the specific value of laser power is required for creating a stable melting pool where all the powder is melted [1,23]. The influence of a laser beam is shown in Figure 2. The transferred energy becomes weaker on the sides of the melting pool [24]. For that reason, Wang Di et al. [25] stated that it is important to set the hatch distance (i.e., the distance between two neighbouring laser paths) in order to create at least a 30 % overlap of the welds. If the hatch distance is not selected correctly, the final layer will not be homogeneous because the scanned lines will not be well connected, which leads to a higher porosity in the part.

The scanning speed is a crucial parameter for reducing the printing time. This parameter should be chosen in a range of values that prevents the melting process from creating defects such as keyholes or balling effect. The keyhole effect may be observed for a combination of a low speed and a high laser power. The material may evaporate during laser exposure and initiate a hole in the layer [1,26]. The balling effect may occur for high scanning speeds and high laser power. The material has no chance to melt and, therefore, it forms balls that are attached to the surface [26,27].

The last parameter is the layer thickness, which is widely chosen from a range of 20 μm to 100 μm according to the required dimension accuracy, the surface roughness [28,29] and ability of the machine to properly melt the chosen layer thickness. The higher layer the thickness, the less time it takes to print a part. Experiments with hybrid layer thicknesses are performed on prints to reduce the fabrication time while preserving the high quality of the outer surfaces [30].

In addition to the above-mentioned primary parameters, the influence of scanning strategy should be also mentioned. A correctly chosen scanning strategy may reduce the porosity, residual stresses, and final mechanical properties of the part [31]. Several strategies for effective scanning of the part cross-section were defined throughout the development of the SLM technology [1]. The three most common strategies are shown in Figure 3 [32]. During this work, the zig-zag strategy with a 90° rotation on each layer was selected. This setting was mentioned in the study by Kurzynowski [33] in which a 99.83% relative density of the material was reported.

### 2.3. H13 Powder Tool Steel

Conventionally manufactured H13 tool steel has a good quenching ability and may reach very high strength (YS = 1650 MPa, UTS = 1990 MPa) [14]. Furthermore, it has great abrasion resistance and high thermal fatigue resistance [34,35]. The powder version of H13 tool steel used for additive manufacturing has different mechanical properties (YS = 835 ± 23 MPa, UTS = 1620 ± 215 MPa) [14,36,37].

A deep investigation of scientific studies was performed to map the SLM technological parameters currently used for manufacturing H13 tool steel. The outcomes are summarised in Table 1. Research teams used various settings of layer thickness, laser power, scanning speed, and hatch distance. The range of process parameters in this study was selected according to the mentioned studies.

### 2.4. Analysis of H13 Powder

Powder that was previously used three times was used in the experiment to fabricate the samples. In the initial part of the work, the particle topology, size distribution, and chemical composition of powder was analysed for qualitative and quantitative comparison of the virgin and used powder. The analysis was performed by taking SEM images using a TESCAN VEGA 3 (TESCAN ORSAY HOLDING, a.s., Brno, Czech Republic) with a 20 kV accelerating voltage and 500× magnification. Figure 4 shows the difference between the virgin and used powders. We assume that the smaller particles which are present in the virgin powder may evaporate during the SLM process. There are no visible differences in the particle topology in either of the studied samples.

The TESCAN VEGA 3 analytical tool and its EDX module were also used to analyse the chemical composition of both the powders. The results presented in Table 2 show little differences between the used and new powders. For new powder, the EDX analysis showed that Cr and Si components do not fit into the limits provided by the manufacturer by 0.1 wt%. Possible reason may be reduced accuracy of this method. Due to the fact that no analysis for precise determination of carbon content was done, we can only presume that the balance of analysis is carbon.

Size distribution was examined with the use of the laser diffraction method on a Bettersize S2 device (Bettersize Instruments Ltd., Dandong, Liaoning, China). The results of the test derived from five measurements for each specimen were distribution curves of the particle sizes as shown in Figure 5.

As observed previously in the SEM analysis of powders, the laser diffraction method verified a higher particle distribution on the used powder with 10 μm larger particles. This was confirmed by a *t*-test which outcome was the level of statistical significance P=0.022. The effect of the size of the difference between the powder particles was also assessed using Cohen’s d, which was 0.33. Based on this value, it is possible to regard this influence as being rather small [41]. The results may also be interpreted as 63% of all the particles from the used sample being bigger than the mean value of the particles from the virgin sample. This means that the small particles were evaporated during the build process or sieved during the sieving process. This rising trend is summarised in Table 3. The powder manufacturer declares that the new powder has a size distribution from 10 μm to 45 μm. Both of the measured samples overlay the maximum value of the range, but it is still possible to use the powder for printing. As a result, there is nothing to prevent the powder from being used.

### 2.5. Design of Experiment

DOE principles were used to prepare the specimens for testing. Specifically, full factorial design defined by Equation (Equation 2) was emloyed. Using this approach, all the possible combinations of factors are tested with further evaluation of the selected responses.
(2)$$k=U^{F}$$
where *k* means the number of specimens, *U* means the number of levels and *F* the means number of factors. For DOE, it is important to apply randomness to prevent systematic or personal errors. Within this work, full factorial DOE was used for 2 factors—laser power and scanning speed—with three levels. Remaining two variables of VED equation, i.e., the layer thickness and the hatch distance were fixed to the values of 0.03 mm and 0.12 mm, respectively. To enhance the statistical part of the experiment, the testing matrix was replicated three times with the specimen position being changed in each replication. The parameters of the DOE are shown in Table 4. Table 5 shows the setting of the process parameters for each specimen.

### 2.6. Production of the Specimens

For the experiment, a cube sample with 10 mm edge length was chosen. All specimens were attached to the build platform with a block support structure so they could be removed later without a wire cutter. The specimens were evenly distributed on the build platform in the pattern sown in Figure 6. A SLM280HL printer with an yb:YAG laser with wavelength λ = 1.064 μm was used to print the specimens. The printing was performed under a N${}_{2}$ atmosphere with 0.03% of oxygen and 200 °C preheated building platform. After each printed replication, a controlled cooling protocol was applied to reduce temperature from 200 °C to 35 °C within 8 h. This protocol was applied based on previous experience with H13 tool steel to prevent thermal shock.

As described above, the testing matrix was replicated three times with variation in the specimen positions. These positions were determined by a random number generator, and the results are shown in Table 6. The number of the specimen from these coordinates is the number of a combination from Table 5.

### 2.7. Analysis of Porosity

In today’s practice, there are three commonly used methods for evaluation of the porosity of metallic parts, i.e., CT scanning, optical metallography (OM), and the Archimedes method. A qualitative comparison of all the three methods is given in the article of Wang et al. [42]. In this work, optical metallography is shows higher precision than CT or the Archimedes method. However, the information level of OM is assessed to be medium due to the fact that only one plane of the specimen is usually evaluated and, therefore, no data about the rest of the specimen are obtained.

With respect to the aim of this work, the specimens were evaluated based on optical microscopy and subsequent image analysis of one metallographic cut. Therefore, the first part of the process was the preparation of the specimens for microscopy using metallographic preparation tools and devices from Struers company (Struers ApS, Ballerup, Copenhagen, Denmark). A Secotom-50 saw with a Struers 30A15 cutting disc was used to cut 2 mm from one of the side faces of each specimen perpendicular to the layering plane (Figure 7a). Then the specimens were mounted in PolyFast resin using Citopress-15 device and polished on a Tegramin 25 device. Details of the grinding and polishing process are shown in Table 7.

The images of polished surfaces were acquired with a Carl Zeiss Imager M2 optical microscope (Carl Zeiss AG, Oberkochen, Germany) with 50× magnification. The images were taken in the automatic mode to cover the whole surface of the specimen and were subsequently combined using ZEN software (Carl Zeiss AG, Oberkochen, Germany), as shown in Figure 7b. NIS-Elements software (NIKON CORPORATION, Konan, Minato-ku, Tokyo, Japan) was used to measure the porosity. Dimensions of 9.6 mm × 9.6 mm were chosen for measuring the porosity to avoid the roughness of the top and bottom surface as shown in Figure 7c. Furthermore, the area where the borderlines are located was removed from the porosity measurement. All the gathered data are shown in Table A1 and in Figure A1–Figure A3, which may be found in the Appendix A.

### 2.8. Analysis of Hardness

The material hardness was measured in the same cut section as the porosity was measured. The experiment was performed on a Struers Duramin-40 device (Struers ApS, Ballerup, Copenhagen, Denmark), which is suitable for measuring Vickers hardness. It was decided to measure the hardness using a 2 × 9 indentation matrix. The test started 1 mm above the support structure plane and rose to the top of the specimen. The testing matrix is shown in Figure 8.

For the experiment, an impact weight of 30 kg was chosen with a 10 s application of the load. The impact diagonal was measured automatically by the microscope that is a part of the Struers Duramin-40 device. For homogeneous specimens, the software had no problem with the measuring distances. If there was a specimen with a higher porosity and the software detects a difference between the diagonal higher than 5%, then the measurement was marked as being invalid.

### 2.9. Theoretical Aspect of the Issue

In this study, laser power *P*[W] and scanning speed *v* [mm/s] were the studied factors of the SLM process. The importance of these parameters is mentioned in equation of VED (see Equation (Equation 1)).

The main measured outputs defining the process quality were porosity and hardness. The porosity is defined as a fraction of air volume to solid volume in the specimen. Assuming that the weight of air is insignificant to the weight of the material, it is possible to state that ($m_{material}\approx m_{specimen}$) [kg]. Porosity ϕ [-] can be described by Equation (Equation 3).
(3)$$\phi \left(VED\right)=\frac{V_{air}}{V_{specimen}}=1-\frac{V_{material}}{V_{specimen}}=1-\frac{\rho_{specimen}}{\rho_{material}}$$
(4)$$\rho_{specimen}=\left[1-\phi \left(VED\right)\right]\rho_{material}$$

Assuming the porosity of the sample is zero percent in the case of an ideal printing setting and the relative density of the sample is close to the relative density of the material ($\rho_{material}=7800$ kg/m${}^{3}$), then Equation (Equation 4) may be used to specify Equation (Equation 5) for the relative density of the material.
(5)$$\lim_{\phi \to 0}\rho_{specimen}=\lim_{\phi \to 0}\left[1-\phi \left(VED\right)\right]=\rho_{material}$$

The main objective was to find the volume of VED to reach the required lowest porosity and satisfactory hardness. Phenomenologically, the dependence of both quantities may be described by Equations (Equation 6) and (Equation 7).
(6)$$\phi \left(VED\right)=K_{1}e^{-\frac{VED}{l_{1}}}$$
(7)$$HV_{30}\left(VED\right)=-K_{2}e^{-\frac{VED}{l_{2}}}+K_{3},\;\{K_{1},K_{2},K_{3},l_{1},l_{2}\}\in R$$

By adjusting both equations, it is possible to describe the dependency between the HV30 hardness and porosity ϕ in Equations (Equation 8) and (Equation 9).
(8)$$HV_{30}\left(\phi \right)=-K_{2}{\left(\frac{\phi }{K_{1}}\right)}^{\frac{l_{1}}{l_{2}}}+K_{3}$$
(9)$$HV_{30}\left(\phi \right)=-A\phi^{b}+K_{3},\;A=K_{1}^{-b}K_{2},\;b=\frac{l_{1}}{l_{2}}$$

In the first approximation of experimental data, the assumption is that the coefficient b=1. In which case, the relationship between the porosity and hardness has a linear trend.

The least squares method was used for an approximation of the data with regard to relationships proposed in Equations (Equation 6), (Equation 7) and (Equation 9). The related phenomenological coefficients were found by numerical calculation. Specifically, the maximal slope method was used.

## 3. Results

Figure 9 shows the dependence of porosity on VED. In the figure, it is possible to see that with rising VED, the porosity decreases. A porosity around 0.2% was reached on specimens with VED from 100.3 J/mm${}^{3}$. Specimens with VED lower than 87.5 J/mm${}^{3}$ show a higher distribution and the area between approximation curves increases. The data were approximated using Equation (Equation 6). The exponential character of porosity-VED dependence induces a higher sensitivity of the extrapolated data in the area of VED ∈ [0; 40] J/mm${}^{3}$ where a low amount of measured data is available. For this reason, a sample with VED = 0 J/mm${}^{3}$ which represents non-melted powder, was added to support the approximation.

Figure 10 shows the influence of VED on hardness. In the figure, it is possible to see that with rising VED the hardness also increases. From the VED value of 87.5 J/mm${}^{3}$, hardness values are more stable and approach 600 HV. For lower energies, the hardness is lower in the case of higher porosity, and it create less resistance to the indenter. The data were approximated using Equation (Equation 7). To extrapolate the influence of hardness on VED, fixed value 0 HV30 for non-melted powder was used.

Mean values of porosity and hardness were also displayed in P-v space (see Figure 11) which is often used for evaluation of process parameters for L-PBF technologies. In our case, this display offers better understanding of our results for wider community.

Figure 12 shows the correlation between hardness and porosity. In this figure, it is possible to see an almost linear trend with decreasing hardness and rising porosity. This finding confirms that it is possible to use coefficient b=1 in Equation (Equation 9). A comparison of coefficients of the approximation and regressive model (see Equation (Equation 9)) shows a good match for both coefficients. Phenomenological Equations (Equation 6) and (Equation 7) may be considered authentic. These findings may also be used for faster porosity prediction.

## 4. Discussion

The first part of the article focuses on the determination of the influence of VED on porosity for the reference H13 tool steel. Specifically, laser power and scanning speed were selected as the two monitored inputs while hatch distance and layer thickness were kept constant. Full factorial DOE was used in this study to systematically test multiple combinations of laser power-scanning speed combinations. For an evaluation of the porosity, the optical metallography (OM) method was employed. The results show that at least 100.2 J/mm${}^{3}$ VED is needed to obtain the minimal values of porosity at a level of 0.2%. The tested combinations with higher VED did not lead to any minimisation of this value. The results published by Laakso et al. [11] show a porosity of 0.07% for specimens built under 102.8 J/mm${}^{3}$ VED. Within their study, they also used the OM method for measuring porosity, but with a lower magnification. This approach may show lower porosity due to it missing the small pores in the specimen. Mazur et al. [38] focused on the same topic and found the best results in the VED range of 80 to 120 J/mm${}^{3}$ with a porosity of 0.12% to 0.01%, respectively. They used the CT method for measuring the porosity. Yonehara et al. [39] used Archimedes methodology to evaluate the porosity of SLM-printed specimens. Assuming a density of 7.78 g/cm${}^{3}$, they reported 0.1% porosity for 79.1 J/mm${}^{3}$ VED. In summary, these studies show that a VED of between 80 and 120 J/mm${}^{3}$ is suitable for obtaining a dense SLM-processed H13 tool steel. Any VED under this range generates higher values of porosity together with a high scatter of the data.

None of these articles considered a replication of their experiments to support the results from a statistical point of view. In terms of the results presented in this work, a high scatter in the porosity data was observed especially for specimens built with lower VED values. From this perspective, it seems inappropriate to formulate a conclusion about porosity from just one observation.

The second part of this article focuses on determining the influence of VED on Vickers hardness. Here, the reported results are in accordance with the findings of other authors for as-built H13 tool steel. Krell [15] measured 607 HV for VED 111 J/mm${}^{3}$ and Ren [43] reached 561 HV for VED 106 J/mm${}^{3}$.

In the last part of the article, the results for porosity and hardness were further analysed and mathematical relations between these quantities and the applied VED were proposed. Both relations were defined as being exponential with an experimental evaluation of the related constants. The last tested dependencies studied in this work was the hardness-porosity relation. Experimental data were fitted using a linear function with a sufficient coefficient of determination. Similar findings are published in a study by Tucho et al. [44], which deals with 316L stainless steel. Here, the researchers investigated a linearly decreasing trend for hardness with rising porosity. The same material for the experiment was used in the work of Cherry et al. [45] study, and their work reported similar findings. The same linear trend was obtained for Co-Cr material by Tonelli [46]. In the currently available literature, no study has been reported for H13 tool steel, therefore, our work fills the gap in this area together with statistical support of the data. In a detailed analysis of the results, none of the mentioned articles describe equations for fitting a trend for porosity-hardness relation. We presented an equation with an accuracy of R${}^{2}$ = 0.93 to confirming the fitting of the equation. This linear trend may be applied to various materials and may be used to reduce time and costs during the preparation of parameters for parts with a required porosity or hardness. Such a model may be also valuable to perform faster prints with higher levels of porosity which may subsequently lowered using Hot Isostatic Pressing (HIP) technology.

Limitations of the presented results may be seen mainly in the fact that only two variables selected from VED equation were studied. In the event that a layer other than 30 μm thick is selected, the recommended range of VED may not be suitable. Another limitation is the low number of specimens for solid statistical support of the presented results. For a better approximation, it would be necessary to increase the number of samples and replications for specimens built with VED lower than 100 J/mm${}^{3}$ because there is a large scatter of data. In future work, our team of authors would like to map this area in more detail to verify our data. Our another goal is to study the microstructure of individual specimens in order to determine mechanisms of porosity formation on different VED levels.

## 5. Conclusions

In this work, we designed phenomenological models to map the relation between the applied volumetric energy density and the resulting porosity and hardness of the material. The main findings and highlights of the study may be summarised as follows:It was confirmed that to obtain material with a minimal possible porosity for H13 tool steel, VED of at least 100.3 J/mm${}^{3}$ must be applied. This setup resulted in a porosity of 0.2%. Higher VED does not lead to any further minimisation of porosity.With respect to our results, both the relations of porosity–VED and hardness–VED may be approximated with anexponential function. For VED lower than 87.5 J/mm${}^{3}$, a high scatter in values of porosity occurs.The relation between hardness and porosity was confirmed to be linear. This assumption was proved by fitting experimental data with a linear regression model with R${}^{2}=0.93$. Such a result may be used for predicting the porosity or hardness by VED.

## Figures and Tables

**Figure 1 materials-14-06052-f001:**
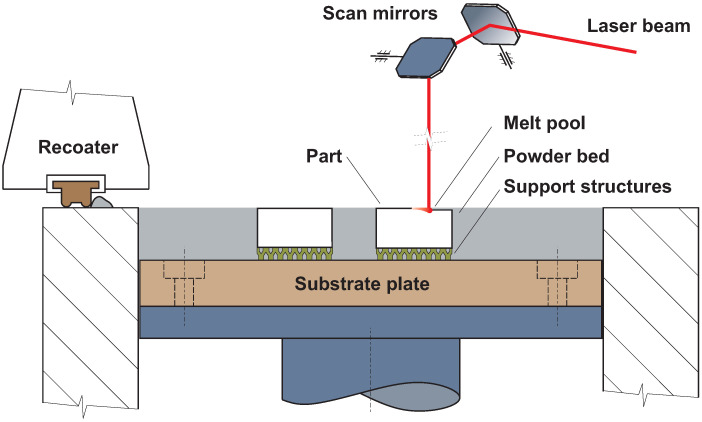
Basic principle of SLM technology.

**Figure 2 materials-14-06052-f002:**
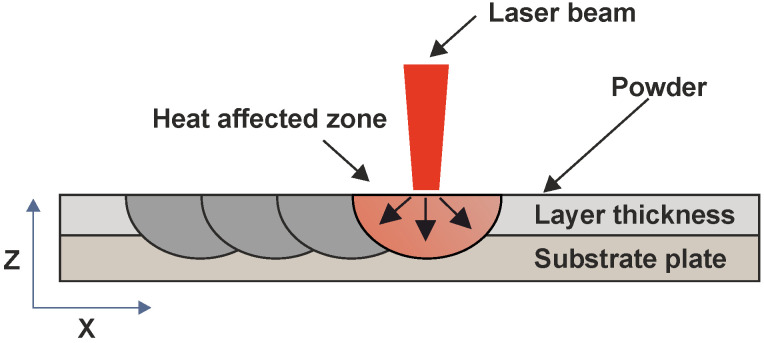
Heat affected zone during melting.Heat affected zone during melting.

**Figure 3 materials-14-06052-f003:**
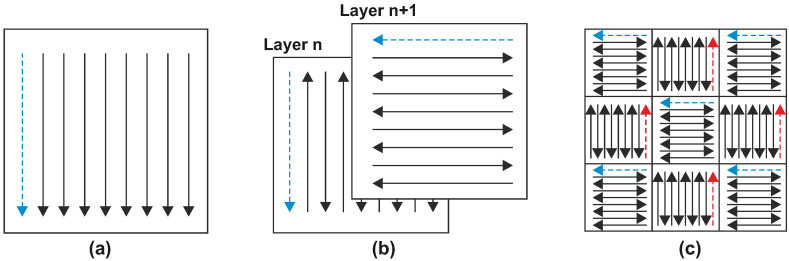
Scanning strategies (**a**) unidirectional, (**b**) stripe hatch with 90° rotation (**c**) chessboard.

**Figure 4 materials-14-06052-f004:**
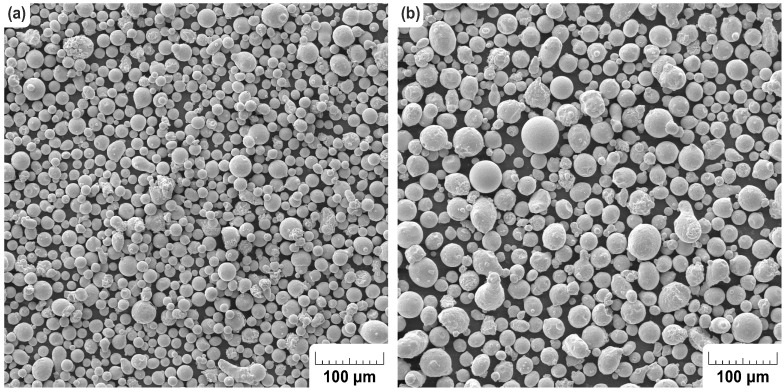
SEM pictures of H13 tool steel powder: (**a**) New, (**b**) Used.

**Figure 5 materials-14-06052-f005:**
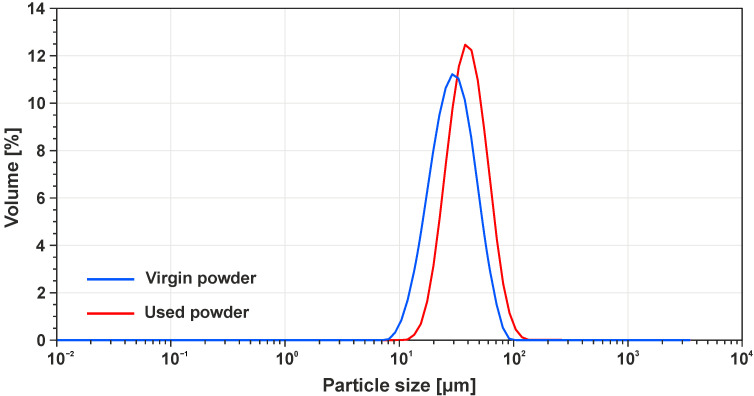
Particle size distribution for H13 powder.

**Figure 6 materials-14-06052-f006:**
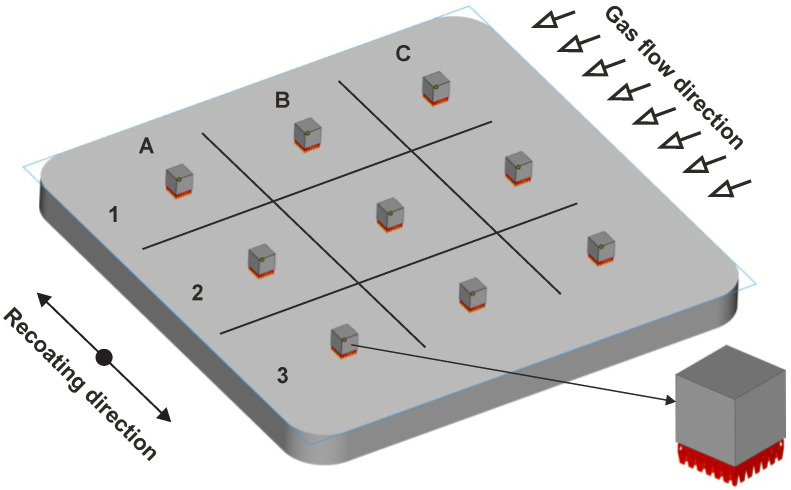
Placement of specimens during printing.

**Figure 7 materials-14-06052-f007:**
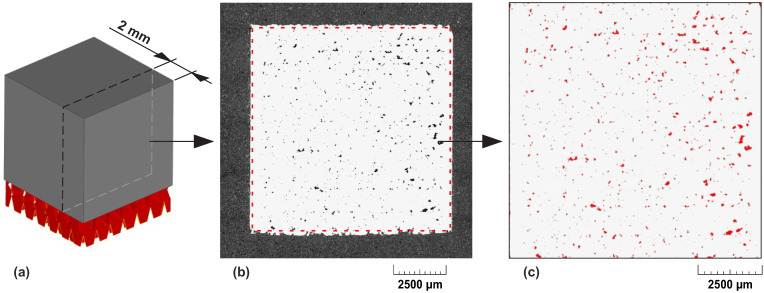
Porosity evaluation process (**a**) Cut section for porosity evaluation (**b**) Picture of the whole specimen cross-section (**c**) Image analysis with pores highlighted in red.

**Figure 8 materials-14-06052-f008:**
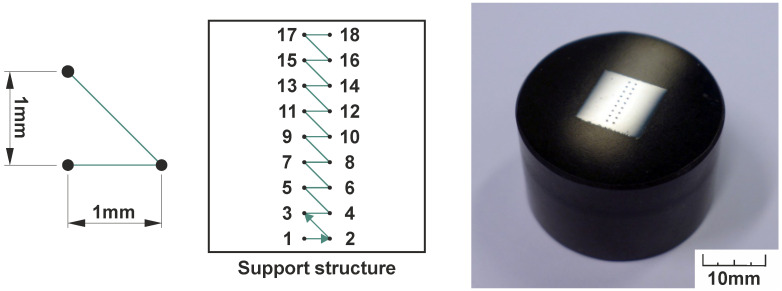
Pattern of hardness measuring.

**Figure 9 materials-14-06052-f009:**
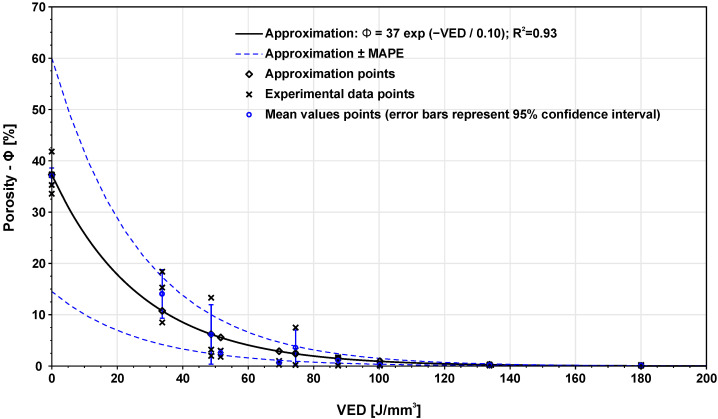
Mathematical relation between porosity and applied VED.

**Figure 10 materials-14-06052-f010:**
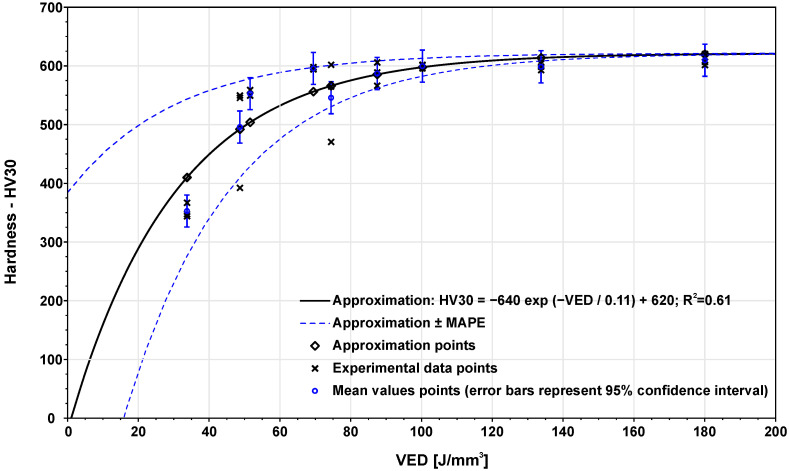
Mathematical relation between HV30 hardness and applied VED.

**Figure 11 materials-14-06052-f011:**
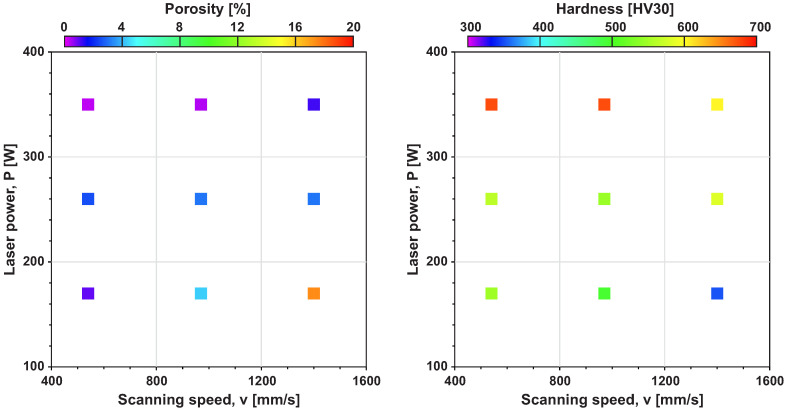
Data of porosity and hardness displayed in P-v space.

**Figure 12 materials-14-06052-f012:**
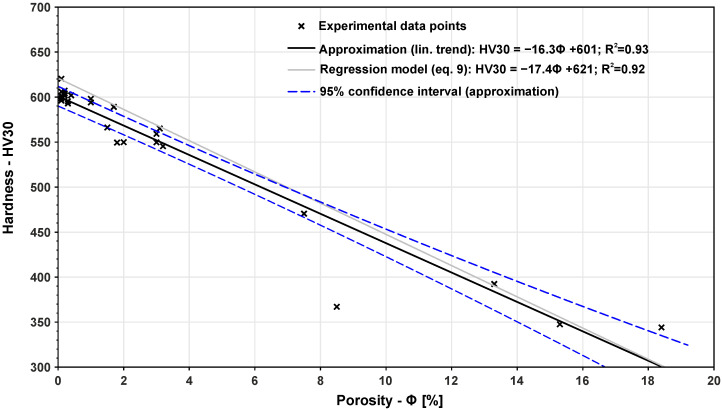
Mathematical relation between HV30 hardness and porosity.

**Table 1 materials-14-06052-t001:** SLM process parameters for H13 tool steel powder.

Reference	SLM	[11]	[15]	[38]	[39]	[40]
Laser power [W]	350	250	100	175	375	280
Scanning speed [mm/s]	1400	994	300	607	790	980
Layer thickness [mm]	0.03	0.03	0.03	0.03	0.05	0.04
Hatch distance [mm]	0.10	0.10	0.10	0.12	0.12	0.12
VED [J/mm${}^{3}$]	84.03	84.03	111	80	79.1	59.51
Porosity [%]	-	0.09	0.50	0.12	0.10	1.40

**Table 2 materials-14-06052-t002:** EDX Chemical composition of H13 powder [wt%].

Element	New Powder	Used Powder	ASTM A681
Fe	90.80	90.90	Balance
C	Balance	Balance	0.32–0.45
Cr	5.60	5.50	4.75–5.50
Mn	0.60	0.60	0.20–0.60
Mo	1.00	1.00	1.10–1.75
Si	0.60	0.80	0.80–1.25
V	1.10	0.90	0.80–1.20

**Table 3 materials-14-06052-t003:** Mean particles size of powder samples.

	D10 [μm]	D50 [μm]	D90 [μm]
H13 Virgin	16.49	29.42	51.13
H13 Used	23.66	39.07	64.48

**Table 4 materials-14-06052-t004:** DOE parameters definition.

Parameter	Value	Parameter Details
Number of factors	F=2	Factor 1: laser power *P* [W]
		Factor 2: scanning speed *v* [mm/s]
Number of levels	U=3	Factor 1: 170 W; 260 W; 350 W
		Factor 2: 540 mm/s; 970 mm/s; 1400 mm/s
Number of specimens for one batch	k=9	
Number of replications	3	
Total specimens	27	

**Table 5 materials-14-06052-t005:** Input parameter values.

Combination	1	2	3	4	5	6	7	8	9
P [W]	170	170	170	260	260	260	350	350	350
v [mm/s]	1400	540	970	1400	540	970	1400	540	970
VED [J/mm${}^{3}$]	33.7	87.5	48.7	51.6	133.7	74.5	69.4	180	100.2

**Table 6 materials-14-06052-t006:** Randomizationed position of specimens.

	Replication 1			Replication 2			Replication 3		
	A	B	C	A	B	C	A	B	C
1	7	4	6	1	3	2	6	1	8
2	1	5	9	4	8	7	2	7	5
3	8	3	2	9	6	5	3	4	9

**Table 7 materials-14-06052-t007:** Grinding parameters.

Step	Grinding Paper (Grit Size)	Force [N]	Time [min]	RPM [min${}^{-1}$]	Grinding Medium
1	SiC 500	30	6	300	Water
2	SiC 1200	25	5	300	Water
3	SiC 2000	20	5	300	Water
4	SiC 4000	15	3	300	Water
5	MD-Nap	15	4.5	150	Dia Duo 1 μm

## Data Availability

The data presented in this study are available on request from the corresponding author.

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
