# Peer review of "Influence of Selective Laser Melting Technology Process Parameters on Porosity and Hardness of AISI H13 Tool Steel: Statistical Approach"

_materials, 2021, doi:10.3390/ma14206052_

Round 1
Reviewer 1 Report
This manuscript presents a study of manufacturing parameter variation for AISI H13 tool steel components (cubes) made via a Powder Bed Fusion process. Power and scan speed were varied and a study of the porosity was performed. Subsequently, Vickers hardness testing was conducted on the samples. Statistical analyses were performed in order to draw out relationships between porosity versus VED, HV30 versus VED, and finally HV30 versus porosity.
There is much to commend about this work. The researchers have done a good job on the basics of sample preparation and optical analysis/hardness testing. I have minor corrections that relate to this aspect of the work towards the end of this review.
However, I do have a major concern around the interpretation of the results and the outcomes of the statistical analysis. The final result in figure 12 seems absolutely reasonable. Hardness should reduce as a result of porosity increase. Given that the hardness is conducted at a relatively high load (30kg), the test volume underneath should contain a good portion of pores when porosity is at higher levels. Thus, porosity should affect the hardness (causing a reduction in hardness). However, the preceding results relating porosity and hardness to VED seem difficult to believe. Yes, the trend is clear from the graph but the physical phenomenon associated with the relationship is not clear to me. This is a common issue in statistical analysis - statistics may show a relationship that can be modelled with a regression analysis, but this regression analysis (and the discussion of the manuscript) does not offer a description of causality (cause and effect) in the physics of the process. The relationships that form the basis for equations 7 and 8 are introduced too casually and are not physically justified in my opinion.
Better frameworks relating parameter selection to defects in AM parts are now widely available. Generally, there are three defective zones in P-v (power, scan speed) space: keyhole, lack of fusion, and balling zones. The optimal processing window is called the conductive zone. Keyhole defects occur as a result of high power, low speed giving laser-beam trapping and high aspect melt pools. Pore formation is related to destabilisation of the moving melt front. Lack of fusion can occur due to lack of penetration or lack of lateral overlap and generally occurs a low power and high speeds. Lack of fusion defects are non circular in appearance. Balling occurs as a result of high power, high speed, and breakdown of the melt due to surface tension. Another source of defect not related to machine parameter selection can be present due to porosity in the initial powder. A recent publication [Gordon, et al. (2020). Additive Manufacturing, 36, 101552] (to which I have no affiliation) gives an excellent overview of the thermodynamic and geometrical relationships that govern the defects in P-v space and due to initial porosity in the powder. The current thinking in the field does not correlate porosity with VED as described by the current manuscript. Indeed, the current work does not vary hatch spacing or layer height and so the correlation to VED in this study is very weak in my opinion. I would strongly recommend that the completely review their work on the basis of P-v diagrams and, in particular, on the Lack of Fusion defect geometrical relationship provided by Tang et al.
Minor points:
The process is usually termed Powder Bed Fusion and it is important to distinguish the energy source as laser (L-PBF is used commonly).
The term 'quasi-static and dynamical' is used. Quasi static is a theoretical condition that cannot be achieved. Simply, use the term static instead of quasi static.
The parameters for balling are high power and high speed. It is said in the manuscript that balling occurs at high scan speeds and low laser power, which I believe to be incorrect.
Figure 3(b) reportedly shows keyhole defects. These do not look like keyhole defects to me, but rather more like lack of fusion defects. The cartesian directions are not indicated in the figure, so it is difficult to have confidence in this image.
Figure 3 (and elsewhere) seems to be taken from other sources. This is poor practice as the figures may be subject to copyright/ownership elsewhere. As a rule generate your own diagrams and figures rather than using someone else's. (even the scale bar formats across figure 3 are different, which looks inconsistent).
on line 147, the manuscript refers to 'The authors...'. Usually this would mean the authors of the current study, but it may refer to authors of other studies. This is confusing. Only use 'the authors' when referring, in third person, to the authors of the current manuscript.
On line 158, change 'may get evaporated' to 'may evaporate'.
Table 2. I appreciate that EDX is not good for measuring Carbon content, but is there another method that can be used to measure the Carbon content? Or are we left to assume that the balance of elemental analysis is Carbon? Some of the constituents (Cr and Si in the new powder) seem to be out of range. No comment is made about this.
Powders were reused three times and a t-test gave a P=0.022. This p value is less than 0.05, which would mean that the powder size distribution has changed statistically. Nevertheless this is ignored. Why?
A DOE method is proposed but the application of the DOE approach is confusing. Equations 2 and 3 relate to full and partial factorial approaches but only the full factorial is used. There is no need to discuss the partial factorial approach if it is not used. Nevertheless, the DOE analysis is not that useful as it is presented. We do not see the main effects or the interactions between the variables through an effects plot (pareto chart of the standardized effects).
On line 211: why mention python program. Is it relevant?
Finally, the best part of the work is the care and attention that has gone into the preparation of the micrographs; however, the names or brands of the equipment is used too frequently. The principle that would usually be applied when describing sample grinding and polishing is to explain the number of steps and the final stage of grinding/polishing medium used. The point is that there should be enough information for a competent operator using appropriate equipment to develop a similar result to that presented in the manuscript. A competent person could use Buehler, ATM, or Struers equipment among many other choices. It really doesn't matter what machines is used, but the final step (diamond grit or alumina suspension with the particle size) should be given. The way that the procedure is currently written is that Struers equipment is preferred, whereas, a competent operator could use any other machine or product to get the same result.
Author Response
Dear reviewer, please find our response in the attached file.

Reviewer 2 Report
The present manuscript investigated the process optimization of the SLM process to manufacture AISI H13 Tool Steel. The manuscript is interesting and useful for the additive manufacturing community especially the processing of high-strength steel. This can be accepted with minor modifications.
- The basic principle of the SML process is well known. So, it is not necessary to describe it again; better to cite it in the literature if required.
- Equation 1 needs a reference if it is not the author’s own proposed one.
- 2 to Fig. 4 do not correspond with the cited references. If the picture is modified, the author should mention that. Also, they should take copyright permission as they are not open access materials.
- The cited articles are different than the H13 steel. How they are relevant to this study.
- Please add a scale bar for all the images.
- Potential defects in the tool steels are porosity, lack of fusion, keyholes, cracks, etc.; especially hot cracks are obvious. So, all the defects are not porosity. Also, lower energy input creates a lack of fusion-type defects. So, please identify the other defects and describe them accordingly in the manuscript.
Author Response

(The authors gave the same response as above.)

Reviewer 3 Report
In this paper, the effect of SLM parameters on the porosity and hardness of AISI H13 tool steel was studied. The current paper is worthy of investigation and organization. The following comments need to be addressed to be accepted for publication.
- It is better to add a nomenclature table containing all the abbreviations.
- The proficiency of the language needs more improvement in the manuscript.
- The introduction part has many details about the methods. I recommend moving it to the materials and methods part.
- The problem statement is not clear in the introduction part. Why is such porosity formed? How to overcome this phenomenon? only by optimizing the processing parameters?
- I recommend studying the microstructure after the performing the SLM at different parameters. The microstructure (including phase analysis) can help in understanding the hardness change as well as the porosity formation.
- logically by increasing the porosity ratio the hardness should by decreased. Not only hardness but also the other mechanical properties. So, it is important to control the porosity formation. In this paper the authors aim to control the hardness by optimizing the SLM process parameters, therefore, I suggest ordering the optimal companions that give acceptable (minimal) ratio of porosity according to the role of minimum is the best. Or it is possible to make a processing map showing the green processing parameters which can give minimum porosity ratio, this can be done by plotting contour maps for example, choosing laser power as X axis and VED as Y axis then porosity % as Z axis. This map can show the feasible regions of laser power and VED for minimal porosity. It is complicated to read that in table A1
Author Response

(The authors gave the same response as above.)

Round 2
Reviewer 1 Report
Thank you for taking the time to consider the comments. The inclusion of Fig 11 is welcomed.
Reviewer 3 Report
Thanks for taking my comments into your consideration. The manuscript can be accepted in the current form